# Towards Understanding Gated Linear Recurrent Neural Networks

## Abstract

Linear Recurrent Neural Networks (RNNs) have attracted attention for their memory and computational efficiency. In particular, gated linear RNNs enable nonlinear transformations through gating mechanisms while still maintaining linear time complexity by removing hidden states from them. However, the impact of the gate mechanisms and such removal of hidden states from them remains unexplored. Here we empirically investigate the impact of these gating mechanisms and find that gate values near zero or one highly depend on hidden states, leading to unintended distribution shifts of gate values when hidden states are removed in gated linear RNNs. Based on our findings, we propose an approach to mitigate the distribution shifts, which empirically improves performance on long-sequence modeling tasks.

## 1 Introduction

Transformers (Vaswani et al., 2017) have become the dominant architecture for sequence modeling. Despite its empirical success, their self-attention mechanism has a fundamental limitation in modeling long sequences because their computational complexity scales quadratically with respect to the sequence length. *Linear recurrent neural networks* (RNNs) have attracted attention because of their memory and computational efficiency as an alternative to transformers. Linear RNNs have linear dependencies on previous hidden states, which enables us to use parallel computation such as convolution, parallel scan algorithms, improving the computational efficiency of RNNs. To date, various linear RNN variants have been proposed (Gu et al., 2022a;b; Smith et al., 2023; Orvieto et al., 2023; Qin et al., 2023), demonstrating the efficacy on modeling long sequences on benchmarks.

*Gating mechanisms* have demonstrated their effectiveness in RNNs for capturing long-range dependencies by enabling selective memory updates and maintaining gradient stability through gates. Specifically, LSTM (Hochreiter & Schmidhuber, 1997) and GRU (Chung et al., 2014) have been successful on sequence modeling tasks. In particular, gated linear RNNs can incorporate *nonlinear* transformations through gating, while it is possible to achieve parallel computation since gate mechanisms are input-dependent (Gu & Dao, 2024; Qin et al., 2023; De et al., 2024; Feng et al., 2024). However, gated linear RNNs remove hidden states from their gate mechanisms to maintain linear recurrence. Since the impact of this removal has not been thoroughly explored, it may lead to undesirable results.

In this paper, we empirically investigate the impact of removing hidden states from the gate mechanisms to understand their dynamics in gated linear RNNs. To better understand their dynamics on gate mechanisms, we use synthetic data to analyze differences in gate mechanisms between gated linear and gated non-linear RNNs. This approach allows us to control experimental conditions and identify differences more clearly than with complex real-world data.

We find that gate values near zero or one highly depend on hidden states, while inputs contribute broadly to gate values across the entire range from zero to one. As a result, a distribution shift of gate values occurs when hidden states are removed from the gates, where the frequency of gate values near zero or one decreases. This distribution shift is not desirable as large gate values close to one are necessary to capture long-term dependencies. To mitigate this distribution shift, we propose a simple approach that applies the Gumbel-Softmax trick (Maddison et al., 2017; Jang et al., 2017; Li et al., 2018) for initialization of bias terms of gates. Experiments show our proposed algorithm improves performance on long-sequence modeling tasks.

Our contributions are summarized as follows:

- We empirically investigate the impact of removing hidden states from gating mechanisms on gated linear RNNs and find that this causes an unintended distribution shift of gate values.
- We introduce a new bias initialization method to mitigate the distribution shift.
- We evaluate our proposed initialization and show its effectiveness compared to various bias initializations on sequence modeling tasks.

## 2 UNDERSTANDING GATING DYNAMICS OF GATED LINEAR RNNS

We empirically investigate the gating dynamics of gated linear RNNs. We first introduce settings of our empirical studies including models and tasks, followed by highlighting the difference of the gate distribution between the gated linear and nonlinear RNNs. By analyzing distributions of gate values, we show that removing hidden states leads to the shift of distributions.

### 2.1 GATED LINEAR AND NONLINEAR RNNS

We introduce linear RNNs and gated linear RNNs. We focus on multi-layer models commonly used in linear RNNs (Orvieto et al., 2023; Qin et al., 2023). Linear RNNs are variants of RNNs which do not have any nonlinear activation functions. Given a sequence $\mathbf{X} = (\mathbf{x}_0, \ldots, \mathbf{x}_t, \ldots, \mathbf{x}_T)$, where its $t$-th element is a $d$-dimensional vector, $\mathbf{x}_t \in \mathbb{R}^d$, hidden states of linear RNNs are computed with linear recurrence as follows:

$$\mathbf{h}_t = \mathbf{W}_{ih}\mathbf{h}_{t-1} + \mathbf{W}_{ix}\mathbf{x}_t, \tag{1}$$

where $\mathbf{W}_{ih}, \mathbf{W}_{ix} \in \mathbb{R}^{d \times d}$ are weight matrices, $\mathbf{h}_t \in \mathbb{R}^d$ is a vector of hidden states, and $\mathbf{h}_{t-1} \in \mathbb{R}^d$ is that of the previous hidden states. *Gated linear RNNs* (Qin et al., 2023; Feng et al., 2024; Gu & Dao, 2024) enable nonlinear transformations through gating mechanisms while maintaining linear recurrence. To focus on the dynamics of gating mechanisms, we use the *minimal* gated linear RNN defined as follows:

$$\mathbf{z}_t = \sigma(\mathbf{W}_{zx}\mathbf{x}_t + \mathbf{b}_z), \tag{2}$$

$$\tilde{\mathbf{x}}_t = \mathbf{W}_{ix}\mathbf{x}_t + \mathbf{b}_h, \tag{3}$$

$$\mathbf{h}_t = \mathbf{z}_t \odot \mathbf{h}_{t-1} + (1 - \mathbf{z}_t) \odot \tilde{\mathbf{x}}_t, \tag{4}$$

where $\mathbf{W}_{zx} \in \mathbb{R}^{d \times d}$ is a weight matrix, $\mathbf{z}_t$ is an update gate, $\sigma(\cdot)$ denotes the sigmoid activation function, $\tilde{\mathbf{x}}_t$ denotes the transformed input, $\mathbf{b}_z, \mathbf{b}_h \in \mathbb{R}^d$ are bias terms, and $\odot$ denotes element-wise multiplication. In the same way, the *minimal gated nonlinear RNN* can be introduced as follows, which includes hidden states in the sigmoid function:

$$\mathbf{z}_t = \sigma(\mathbf{W}_{zx}\mathbf{x}_t + \mathbf{W}_{zh}\mathbf{h}_{t-1} + \mathbf{b}_z), \tag{5}$$

$$\tilde{\mathbf{x}}_t = \mathbf{W}_{ix}\mathbf{x}_t + \mathbf{b}_h, \tag{6}$$

$$\mathbf{h}_t = \mathbf{z}_t \odot \mathbf{h}_{t-1} + (1 - \mathbf{z}_t) \odot \tilde{\mathbf{x}}_t, \tag{7}$$

where $\mathbf{W}_{zh} \in \mathbb{R}^{d \times d}$ is a weight matrix.

### 2.2 SYNTHETIC TASKS

We use two synthetic tasks for sequence modeling. In these synthetic tasks, we used 3-layer RNNs, where the size of hidden states was set to 128, and the dimension of embedding was set to 128. We trained models for 50K steps using AadmW optimizer with a learning rate of 0.0001 and a batch size of 64.

The copying task (Arjovsky et al., 2016) aims to evaluate the memorization ability to capture long-term dependencies. The input sequence consists of $M$ memorizing tokens, $T$ dummy tokens, and $M$ output tokens. The task requires to memorize the first $M$ tokens and output the same of the first $M$ tokens on the last $M$ steps. From $N$ tokens $(0, 1, \ldots, N-1)$, the first $M$ tokens are randomly set from $(1, \ldots, N-2)$, the middle $T$ dummy tokens are set to 0, and the last $M$ output tokens are set to $N-1$. We set $N = 10$, $M = 10$, and $T = 500$, resulting in total sequence length 520.

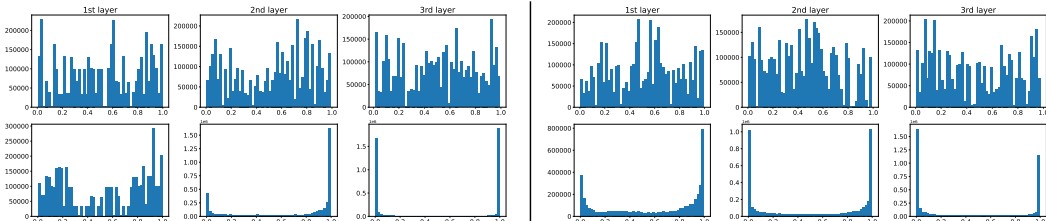

Figure 1: Gate distributions of the gated linear RNN (left panel) and the gated nonlinear RNN (right panel) across layers on the copying task. The top row shows distributions before training, and bottom row shows those after training.

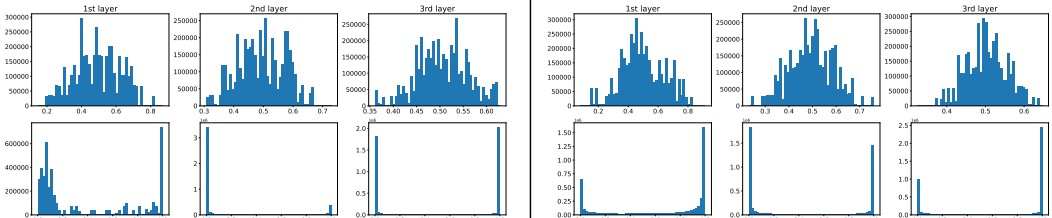

Figure 2: Gate distributions of the gated linear RNN (left panel) and the gated nonlinear RNN (right panel) across layers on the selective copying task. The top row shows distributions before training, and bottom row shows those after training.

Following (Gu et al., 2020), we compute the loss for the output tokens and use UGI (Gu et al., 2020) which initializes biases whose post-activation values follow a uniform distribution.

The selective copying task (Jing et al., 2019; Gu & Dao, 2024) is a variant of the copying task. It requires memorizing tokens at varying positions, while the copying task requires memorizing the first $M$ tokens. For considering the effect of bias initialization, we used standard initialization for bias, which is a sampling from a uniform distribution $\mathcal{U}[-1/\sqrt{d}, 1/\sqrt{d}]$ in this task. We set $N = 16, M = 16$, and $T = 500$, resulting in total sequence length 532.

### 2.3 Empirical Observations of Gating Distributions

We empirically compare gate distributions between the minimal gated linear RNN and the minimal gated nonlinear RNN.

Figure 1 shows gate distributions of both models on the copying task. Before training, these distributions range from zero to one in all layers due to the UGI initialization. After training, distributions largely change for both models, and gate values are concentrated at zero or one in the second and the third layers. However, the trend is different for the first layer; gate values of the minimal gated linear RNN are broadly distributed across the range from zero to one, while those of the minimal gated nonlinear RNN are concentrated at zero or one.

Next, we examine gate distributions on the selective copying task. Results are shown in Figure 2. Before training, gate distributions of both models are distributed around 0.5 in all layers due to the standard initialization unlike UGI distribution. After training, gate values are again concentrated at zero or one in the second and the third layers for both models. Similar to the copying task, this concentration effect is not strong for the first layer of the minimal gated linear RNN, while the concentration also happens in the minimal gated nonlinear RNN.

### 2.4 Gate Distribution Shift

Next, to examine the effect of hidden states, we illustrate and compare gate values $\mathbf{z}_t$, pre-activation values from inputs $\mathbf{W}_{zx}\mathbf{x}_t$ and hidden states $\mathbf{W}_{zh}\mathbf{h}_{t-1}$ of the minimal gated nonlinear RNN given in equation 5, and post-activation values through the sigmoid function as shown in Figure 3. We

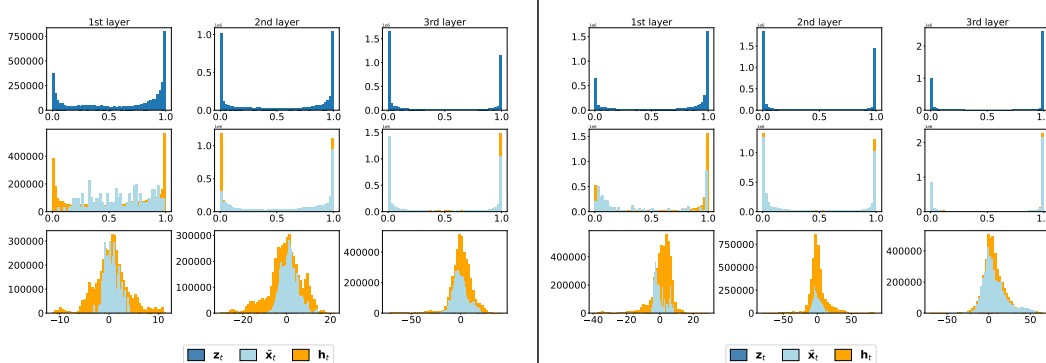

Figure 3: Gate distributions for each layer after training (top), distributions after sigmoid activation for transformed hidden states and inputs after training (middle), and distributions before sigmoid activation for transformed hidden states and inputs after training (bottom). The left panel is for the copying task and right panel is for the selective copying task. Colors mean gate distributions (blue), distributions of hidden states (orange), and distributions of inputs (light blue).

can see that hidden states after sigmoid activation (colored in orange) are distributed close to zero or one. In contrast, inputs after sigmoid activation (colored in light blue) are broadly distributed across the range from zero to one. This distribution shift occurs only in the first layer as layers except for the first layer mitigate the distribution shift due to implicitly containing hidden states from previous layers through gates.

These results indicate that the distribution shift occurs when we remove hidden states. This could be attributed to the information gap between hidden states and inputs. Hidden states have sufficient information for learning parameters on the gate due to preserving information until the $(t-1)$-th step. In contrast, inputs do not have sufficient information because they have information at only current $t$-th step. Since gated linear RNNs cannot use hidden states into their gate mechanisms to maintain linear recurrence, this distribution shift is likely to occur implicitly, which may lead to undesirable outcomes.

## 3 MITIGATING GATE DISTRIBUTION SHIFT BY GUMBEL-SOFTMAX TRICK FOR BIAS INITIALIZATION

To mitigate the distribution shift caused by removal of hidden states in gated linear RNNs, we propose to apply the Gumbel-Softmax trick (Li et al., 2018) to bias initialization.

### 3.1 BIAS INITIALIZATION WITH GUMBEL-SOFTMAX TRICK

The *Gumbel-Softmax trick* (Maddison et al., 2017; Jang et al., 2017), leveraged by Li et al. (2018), is given as follows:

$$G(\alpha, \tau) = \sigma \left( \frac{\alpha + \log(u) - \log(1-u)}{\tau} \right),$$ (8)

where $\alpha \in \mathbb{R}$ is a parameter, $\tau > 0$ is a temperature parameter, $u \sim \mathcal{U}[0,1]$ is sampled from the uniform distribution. We apply this trick to bias initialization of the first layer in equation 2; that is, the bias term $b_z$ is sampled as

$$b_z \sim \sigma^{-1} G(\alpha, \tau).$$ (9)

Following Gu et al. (2020), we sample $u$ from $\mathcal{U}[1/d, 1-1/d]$ instead of $\mathcal{U}[0,1]$ for numerical stability to avoid $\log(0) = -\inf$. Using the above Gumbel-Softmax trick, we can initialize gate biases whose post-activation values become close to zero or one, which is crucial as hidden states contribute to gate values near zero or one. We can therefore expect that the unintended distribution shit we have observed in Figure 3 can be avoided through this initialization. Moreover, we can optimize the balance between zero and one through $\tau$ and $\alpha$ while maintaining a bimodal distribution.

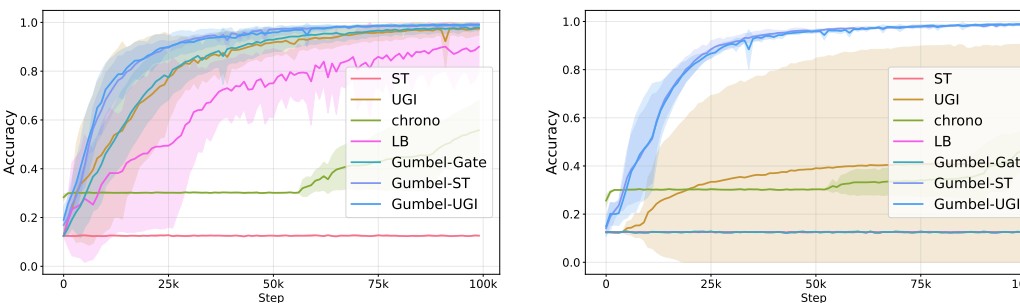

Figure 4: Performances on the copying task. The dummy sequence length is 1000 (left) or 2000 (right).

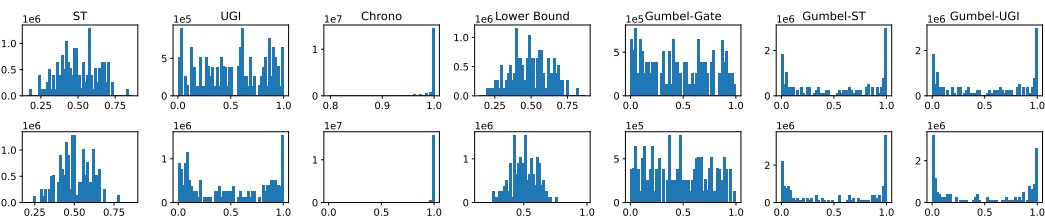

Figure 5: Gate distributions before (top) and after (bottom) training for each method in the first layer on the copying task with the dummy sequence length 2000

Note that we apply this initialization to only the first layer as the distribution shift occurs only in the first layer as discussed in Section 2.4.

Gumbel-Softmax trick for bias initialization is a generalization of the UGI (Gu et al., 2020), which corresponds to the case when $\alpha = 0.0$ and $\tau = 1.0$ in the Gumbel-Softmax trick. We can see this from the definition since $\sigma^{-1}(x) = \text{logit}(x) = \log\big(x/(1-x)\big)$ and it follows that

$$G(0,1) = \sigma(\log u - \log(1-u)) = \sigma\Big(\log\big(\frac{u}{1-u}\big)\Big) = \sigma(\text{logit}(u)) = u. \tag{10}$$

### 3.2 PARAMETER SETTING OF GUMBEL-SOFTMAX INITIALIZATION

We discuss how to set two parameters $\tau$ and $\alpha$ in the Gumbel-Softmax trick. When $\tau$ is 0, the resulting distribution can be considered as an approximation of Bernoulli distribution (Li et al., 2018), where the distribution of gate values is sharp. Moreover, when $\tau$ and $\alpha$ are 1.0 and 0.0, respectively, the distribution can be considered as an approximation of the Uniform distribution. We therefore use $\tau = 0.5$ as a default setting in our proposed approach because $0.5$ can moderately optimize the distribution of gate values through the parameter $\alpha$. We will validate the effect of $\tau$ in experiments.

Regarding $\alpha$, it pushes gate values towards one as it increases. Large gate values are essential for modeling long-sequence tasks. However, when gate values are extremely concentrated at one, hidden states are not properly updated. When $\alpha$ is 0.0 and $\tau$ is 0.5, most gate values are distributed around at zero or one. Therefore, the model with $(\alpha, \tau) = (0.0, 0.5)$ is expected to capture long-term dependencies while maintaining the ability to memorize new inputs. Therefore we use $\alpha = 0.0$ in our proposed initialization. We will validate the impact of $\alpha$ in experiments.

## 4 EMPIRICAL EVALUATION OF OUR PROPOSAL: BIAS INITIALIZATION WITH GUMBEL-SOFTMAX TRICK

We evaluate our proposed approach (denoted as "Gumbel" in figures and tables) in practical settings and datasets. In following tasks, we use 6 layer models, which are also used in literature (Orvieto

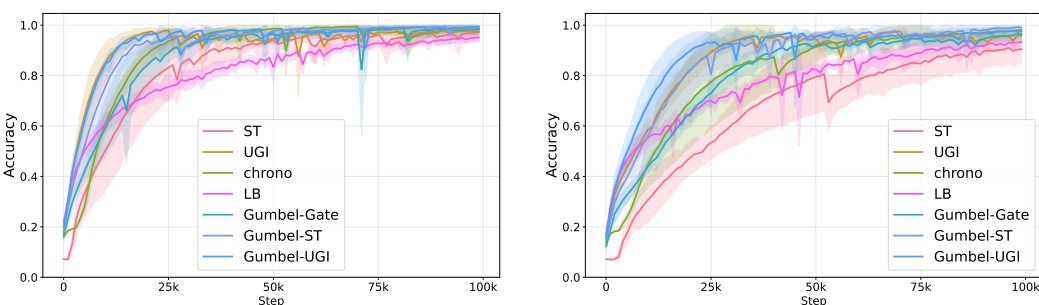

Figure 6: Performances on the selective copying task. The dummy sequence length is 1000 (left) or 2000 (right).

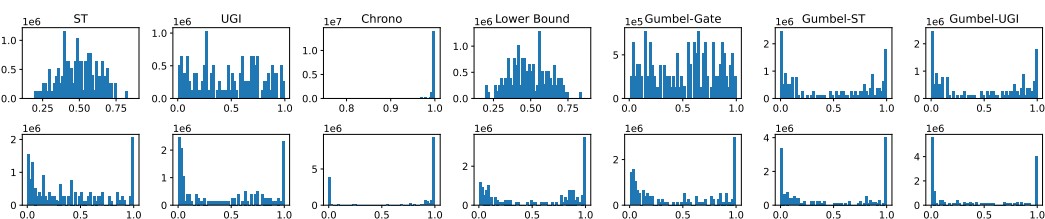

Figure 7: Gate distributions before (top) and after (bottom) training for each method in the first layer on the selective copying task with the dummy sequence length 2000.

et al., 2023; Qin et al., 2023). We compare our proposed initialization with standard initialization (ST), chrono initialization (Tallec & Ollivier, 2018), and UGI (Gu et al., 2020). The parameter $T_{max}$ of chrono initialization was set to the whole sequence length. We also compare gate modifications, including lower bounds (LBs) for gate values (Qin et al., 2023) and Gumbel-Gate (Li et al., 2018) which pushes gate values towards zero or one. We use PyTorch's default parameters[1] for LB without additional initialization because we found improved performance in our experiments. We apply Gumbel-Gate to the first layer to validate the effect of applying bias initialization in the first layer. Following Li et al. (2018), we set the parameter $\tau$ of Gumbel-Gate to 0.9.

### 4.1 SYNTHETIC TASKS

To evaluate the ability of capturing long-term dependencies, we extend the dummy sequence length to 1000 and 2000. We trained models for 100K steps using the same settings as in Section 2.2. In synthetic tasks, we applied Gumbel-Softmax trick to the first layer and applied two initializations to other layers: standard initialization (Gumbel-ST) and UGI (Gumbel-UGI). We also applied Gumbel-Gate to the first layer and standard initialization to other layers. We conducted experiments with three different random seeds.

**Copying task.** Figure 4 shows the mean accuracies and standard deviation across 3 random seeds on the copying task for various initializations and methods, and Figure 5 shows gate distributions under various initializations and methods with the dummy sequence length $N$ of 2000. Our proposed initializations achieved optimal performances at both 1000 length and 2000 length. Although UGI and Gumbel-Gate achieved optimal performances at the dummy length 1000, their performance degraded when the dummy length is 2000. This is why their gate values are broadly distributed across the range zero or one unlike our proposed initializations as shown in Figure 5. Chrono initialization requires many steps to increase accuracy, and chrono initialized biases are concentrated at one across all layers as shown in Figure 5. This suggests that gates values under chrono initialization in all layers are too high to update hidden states, which does not occur in our proposed methods. Regarding LB, it achieved near optimal performances at the dummy length 1000, which suggests that

---

[1]https://docs.pytorch.org/docs/stable/generated/torch.nn.parameter.Parameter.html

lower bounding of gate values enables the model to capture long-term dependencies compared to ST. However, convergence of LB is slow because it struggles to optimize parameters when initializations do not works well, resulting in the degraded performance at the dummy length 2000.

**Selective copying.** Figure 6 and 7 shows the mean accuracies and standard deviation across 3 random seeds and gate distributions, respectively, on the selective copying task for various models with the dummy sequence length $N$ of 2000. All models achieved near optimal performance when the dummy length is 1000, and our initializations and UGI and chrono initialization achieved near optimal performances at both length of 1000 and 2000. Unlike the copying task, this task also requires to capture multi-timescale dependencies because of varying memorizing tokens. According to this requirement, gated values of our proposed initializations are also distributed near zero in addition to one compared to the copying task as shown in Figure 7. Additionally, standard initialization achieved better performances than the copying task. UGI achieved optimal performances at both length of 1000 and 2000 because initialized biases are widely distributed between zero and one. Since UGI can optimize gate values compared to the copying task, the gate distribution became more binary-like distribution as shown in Figure 7. LB also improved performances at both length of 1000 and 2000 compared to the copying task. However, the convergence of LB is slow, which is the same trend with the copying task. Gumbel-Gate achieved nearly optimal performance, while it was not better performance than our proposed initializations because gate values are widely distributed between zero and one unlike our proposed approaches as shown in Figure 7. From these results, our proposed initializations, which encourage a binary-like distribution of gate values in the first layer, are shown to be effective for long-sequence modeling.

## 4.2 LANGUAGE MODELING

Next we evaluate our proposed initialization with a language modeling task on the WikiText-103 dataset. This task evaluates the ability of capturing multi-timescale dependencies. Models are trained for 50K using AdamW optimizer using an inverse-square-root learning rate scheduler with a peak learning rate of $0.0005$, warmup-steps of $4000$, betas of $(0.9, 0.98)$, weight decay of $0.0$, context length of $2048$, and a batch size of 16. The size of hidden states was set to $2048$ with drop rate of $0.1$, and the dimension of embedding was set to $2048$ with drop rate of $0.3$. We use the GPT-2 tokenizer from the Hugging Face Library (Wolf et al., 2020). In the language modeling task, we applied Gumbel-Softmax trick and Gumbel-Gate to the first layer and UGI to other layers.

Table 1: Performances measured by the perplexity for the language modeling task on the WikiText-103 dataset (lower is better). Chrono initialization failed to converge because the loss function continuously increased after 7K steps.

| Method | Valid | Test |
|---|---|---|
| ST | 35.94 | 36.99 |
| Chrono | – | – |
| UGI | 33.87 | 34.78 |
| LB | 40.19 | 41.31 |
| Gumbel-Gate | 33.74 | 35.06 |
| Gumbel-UGI (proposed) | 33.93 | 34.70 |

Table 1 shows performances on the WikiText-103 dataset, measured by the perplexity (lower is better). Our proposed initialization has comparable performance to UGI and Gumbel-Gate, which demonstrates that our proposal is also able to capture multi-timescale dependencies while maintaining the ability of capturing long-term dependencies. Also, our model mitigate overfitting compared to UGI and Gumbel-Gate because our gate values are more close to zero or one, resulting in a flatter loss surface. These results are consistent with the previous study (Li et al., 2018). Chrono initialization failed to converge because the loss function continuously increased after 7K steps. This may be attributed to high initial biases. LB is not better than ST due to overfitting to training data. This could be caused by additional parameters of lower bounds.

## 4.3 SENSITIVITY ANALYSIS

We analyze the sensitivity of the parameters $\alpha$ and $\tau$ used in our proposed bias initialization. In the sensitivity analysis, we applied Gumbel-Softmax trick to the first layer and standard initialization to other layers.

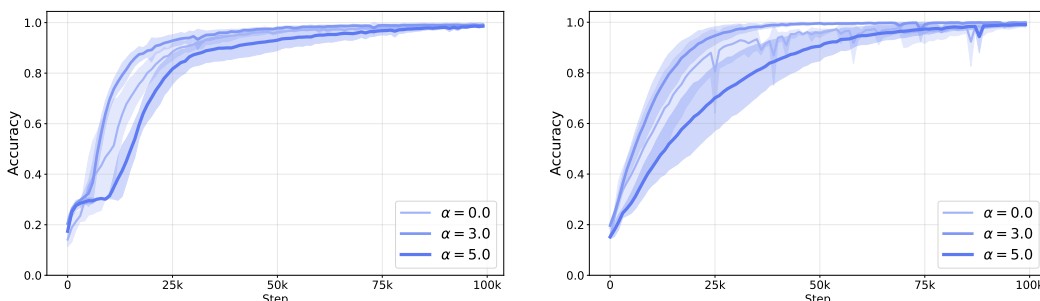

Figure 8: Performances when $\alpha$ varies on the copying task (left) and the selective copying task (right). The length of dummy sequences is 2000 in both tasks.

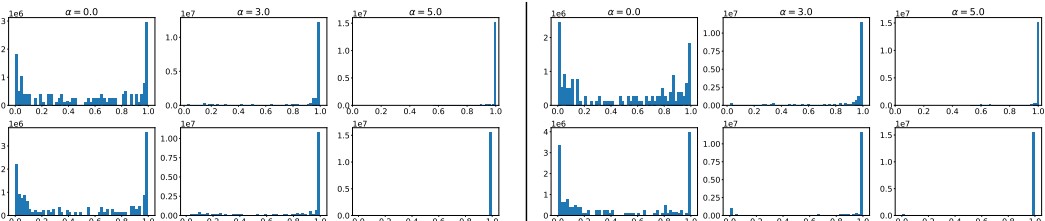

Figure 9: Gate distributions at the first layer under different $alpha$ for the copying task (left panel) and the selective copying task (right panel) with the dummy sequence length 2000.

First, we examine the effect of $\alpha$. Figure 8 shows the mean accuracies and standard deviation across 3 random seeds for three settings of $\alpha \in \{0.0, 3.0, 5.0\}$ on the copying and selective copying tasks, respectively. Moreover, Figure 9 illustrates gate distributions for the same settings of $\alpha$ on both tasks. As we can see, every $\alpha$ eventually achieves the optimal performance in both tasks. However, the convergence with $\alpha = 5.0$ is slower in both tasks because gated values are too high to store new inputs as shown in Figure9.

Next, we primarily consider $\tau$ by comparing combinations varying $\tau$ and $\alpha$. Figure 10 shows the mean accuracies and standard deviation across 3 random seeds on varying $(\tau, \alpha)$ on the copying and selective copying tasks, and Figure 11 shows gate distributions of various initializations and methods on the copying task at the dummy sequence length $N = 2000$. When $\tau$ is 0.2, gate values are extremely concentrated at zero or one as shown in Figure 10. Therefore, it is difficult to adjust the distribution of gate values through $\alpha$. For example, in the copying task, the model cannot solve this task regardless of whether $\alpha$ is 0.0 or 3.0. In contrast, when $\tau$ is 0.9, the distribution differs significantly between 0.0 and 3.0 as shown in Figure 11. This makes it difficult to choose the best parameter because the distribution is sensitive to $\alpha$. For example, in the copying task, the model with $(\tau, \alpha) = (0.9, 0.0)$ cannot solve the task while the model with $(\tau, \alpha) = (0.9, 3.0)$ achieves the optimal performance.

## 5 RELATED WORK

### 5.1 LINEAR RECURRENT MODELS

Linear recurrent models have achieved competitive performance as an alternative to transformers on modeling long-sequence tasks. Many variants of linear RNNs, including GILR (Martin & Cundy, 2018), S4 (Gu et al., 2022a), S5 (Smith et al., 2023), LRU (Orvieto et al., 2023), Griffin (De et al., 2024), and HGRN (Qin et al., 2023), have proven to be able to capture long-term dependencies. Recently, new types of linear recurrent models (Katharopoulos et al., 2020) emerged based on linear attentions. They address the limitation of the expressiveness of linear RNNs. Several studies have explored gated linear recurrent models with linear attention, such as DeltaNet(Schlag et al., 2021; Yang et al., 2024b), Gated Linear Attention (Yang et al., 2024a), Mamba2 (Dao & Gu, 2024),

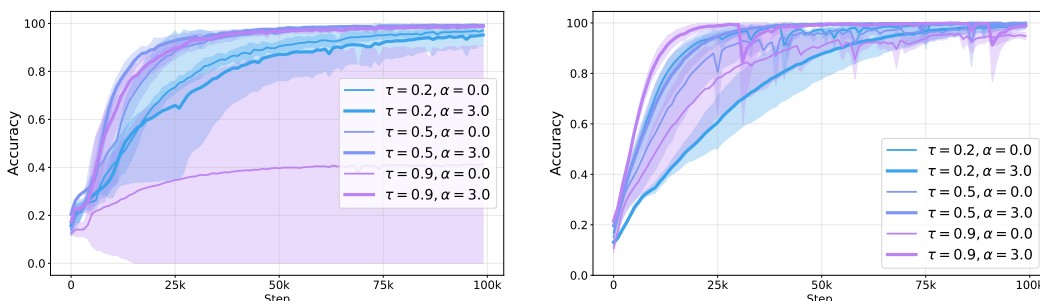

Figure 10: Performances when $(\tau, \alpha)$ varies on the copying task (left) and the selective copying task (right). The length of dummy sequences is 2000 in both tasks.

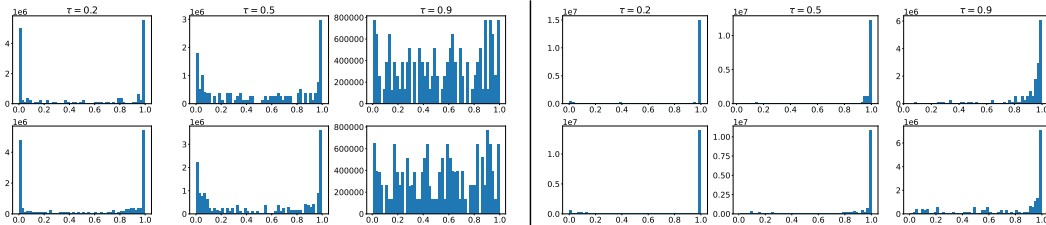

Figure 11: Gate distributions at the first layer under different $\tau$ with $\alpha = 0.0$ (left) and $\alpha = 3.0$ (right) for the copying task with the dummy sequence length 2000.

HGRN2 (Qin et al., 2024), Gated Delta Network (Yang et al., 2025), and RWKV (Peng et al., 2024; 2025), which have demonstrated competitive performance on modeling long-sequence tasks. In the context of these linear RNN studies, we focus on optimizing their gate mechanisms.

## 5.2 OPTIMIZATION FOR GATE VALUES OF RNNS

Efficient optimization for gate mechanisms is crucial for RNNs. Standard initialization struggles to optimize gate values because gradients through gates vanish when gate values are close to zero or one. In nonlinear RNNs, Tallec & Ollivier (2018) introduced a bias initialization so that gate values become high values to capture long-term dependencies. Li et al. (2018) introduced Gumbel-Gate which pushes gate values towards zero or one by replacing the sigmoid activation function with the Gumbel-Gate. Gu et al. (2020) introduced a combination of the gate mechanism and bias initialization to push gate values close to zero or one. ON-LSTM (Shen et al., 2019) explicitly orders neurons to capture dependencies with different time scales. In linear RNNs, HGRN (Qin et al., 2023) is inspired by ON-LSTM (Shen et al., 2019) and introduces lower bounds for gate values. The lower bound increases monotonically from input to output layers, enabling storing information with different time scales at different layers. However, previous studies have not unrevealed the impact of removing hidden states from gate mechanisms in linear RNNs. We investigate this impact and explore effective optimization for their gate mechanisms.

## 6 CONCLUSION

In this paper, we have empirically investigated the gate dynamics of gated linear RNNs. In particular, we have analyzed the impact of removing hidden states from their gate mechanisms. We have shown that distribution shifts unintentionally occur due to such removal of the hidden states. To mitigate this issue, we have introduced a simple way of applying the Gumbel-Softmax trick to bias initialization without requiring architectural changes. Experimental results show that our proposed initialization significantly improves performance on long-sequence modeling tasks. These results suggest that not only architecture design but also the optimization for gate mechanisms is crucial in gated linear RNNs.

**LLM usage:** We used LLMs to polish our texts to choose suitable words and correct grammars.

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
