# OpenReview forum: "Towards Understanding Gated Linear Recurrent Neural Networks"
_ICLR.cc/2026/Conference — Submitted to ICLR 2026_

### Official Review · Reviewer_wM4S · 2025-10-27

**Soundness:** 2
**Presentation:** 3
**Contribution:** 2
**Rating:** 4
**Confidence:** 3

**Summary:**

This paper discusses the compromise in gated linear RNNs, which remove the dependency of the gate projection on the previous hidden state to maintain linear complexity. While it's intuitive that removing this dependency leads to a performance drop, the paper provides a new perspective by analyzing the distribution shift of gate values. The authors uncover that after removing this dependency, the first layer of the model shows a significant distribution shift in its gate values, while subsequent layers exhibit a more moderate shift. Building on this, the paper introduces a trick that applies Gumbel-Softmax initialization to the first layer, forcing the model to output gate values close to either 0 or 1, thus enhancing performance. The experiments verify that this trick is effective for the synthetic recall tasks designed by the authors. However, for more complex language tasks, the trick is not as effective.

**Strengths:**

1. The paper investigates the intuitive performance drop caused by removing the gate's dependency on $h_{t-1}$. It empirically finds that this removal causes a "distribution shift" in the gate values. The discovery that $h_{t-1}$ is the primary driver for pushing gate values towards 0 or 1 in nonlinear RNNs is useful and inspirational.
2. The solution of applying Gumbel-Softmax initialization only to the first layer based on the diagnosis that this is where the shift is most clear, which is a targeted approach.

**Weaknesses:**

1. While the Gumbel-Softmax initialization shows outstanding performance on synthetic tasks, it demonstrates almost no improvement on the more important, complex real-world task of WikiText-103 language modeling (Table 1).
2. All analyses and experiments are conducted on a minimal gated linear RNN or a simple 6-layer version. The paper completely lacks validation on any modern, state-of-the-art Linear RNN architecture (e.g., Mamba, RWKV). And the research can be expanded to modern architectures that adopt the matrix-formed memory (e.g., Gated Deltanet, TTT).
3. The benchmark is relatively insufficient. Some basic widely used language tasks such PIQA, ARC are not included. This may due to that model is too small that cannot handle even a bit complex tasks. However, the results on small models may not be persuasive.

**Questions:**

1. The method excels on copying tasks but shows almost no improvement on WikiText-103 language modeling . Does this imply the method fails to generalize to complex, real-world tasks?
2. The analysis uses a minimal model. How do we know this distribution shift problem even exists in more complex SOTA architectures like Mamba or GLA, which were not tested?

---

> ### Author Response · Authors · 2025-11-23
>
> We appreciate your constructive feedback.
>
> W1/Q1:
>
> Since Language Modeling (LM) tasks require focusing on important parts of input data, input-adaptive mechanisms like gate mechanisms are known to be suitable for LM tasks.
> Therefore, our experimental results on WikTtext-103, where the baselines achieved performance comparable to ours, are consistent with our statement.
>
> In contrast, long sequence modeling tasks require pushing gate values towards one to keep previous information regardless of input data and gate mechanisms struggle to memorize tokens under such a situation.
> Our experiments show that the performances of baselines are indeed suboptimal in the copying and the selective copying tasks when the length of the dummy sequence increases from 1000 to 2000.
>
> W2/Q2:
>
> We would like to clarify that the gate distribution shift occurs caused by the removal of hidden states, and it remains for modern gated linear RNNs such as Mamba and RWKV because they cannot also use previous information through hidden states at the first layer.
> This suggests that modern gated linear RNNs are still likely to struggle to push gated values close to one.
> The HGRN2 paper [1] reported that Mamba underperforms compared to linear RNNs such as S4 and S5 on long-sequence modeling tasks.
> We believe that leveraging our work to modern gated linear RNNs could help identify the underlying causes and improve the ability to capture long-term dependencies.
>
> W3:
>
> Our primary goal is to understand the effect of hidden states on the gate mechanisms, rather than providing state-of-the-art performance on benchmark tasks.
> Our findings can be even more critical for modeling real-world data because they are potentially more complex and large-scale, which makes pushing the gate values towards one more difficult.
> Our experiments on WikiText-103 verifies the versatility of our proposed method, and those on synthetic data show improvement of capturing long-term dependencies.
> Scaling to large models remains future work.
>
> References:
> [1] Zhen Qin, Songlin Yang, Weixuan Sun, Xuyang Shen, Dong Li, Weigao Sun, and Yiran Zhong. HGRN2: Gated linear RNNs with state expansion. In Proceedings of the Conference on Language Modeling, 2024.

---

### Official Review · Reviewer_vbjC · 2025-10-30

**Soundness:** 2
**Presentation:** 2
**Contribution:** 2
**Rating:** 2
**Confidence:** 3

**Summary:**

This paper investigates the gating dynamics of gated linear RNNs, focusing on the impact of removing hidden states from the gate computation. The authors propose a bias initialization method based on the Gumbel-Softmax trick. The proposed method is evaluated on synthetic and real-world sequence modeling tasks with improvements over standard and prior initialization strategies.

**Strengths:**

The proposed method is somewhat clear and easy to follow.

**Weaknesses:**

However, I think the draft has severe drawbacks:

1. I think the main claim of this paper is somewhat not very convincing. The author uses synthetic tasks to show the findings, and only performs it with 3/6 layer models. I don't think that is convincing. For real-world results, the author uses the WikiText-103 dataset with 2048 sequence length, which is somewhat too short in the recent LLM era. Will the proposed findings still occurs in larger models? If the model/data got deeper/larger, will the distribution shift be solved by scaling? Since the results with the proposed method on WikiText-103 is similar to previous results, I don't think it may become a critical issue.

2. The proposed method is somehow not novel. It just changes the bias initialization method with Gumbel-Softmax, which has occurred in previous methods. The change is minor, while the performance gain over previous methods is also minor.

**Questions:**

I think the author should revise the draft to enhance the importance of solving so-called distribution in toy-data and toy datasets.

---

> ### Author Response · Authors · 2025-11-23
>
> We thank you for your effort in reviewing our paper and variable feedback.
>
> W1:
>
> Our key finding is that, even in simple synthetic tasks, gated values fail to approach zero or one when initialization is not appropriate, which is not desirable because large gate values close to one are necessary to capture long-term dependencies.
> It is critical for real world applications because the higher complexity of real-world data makes it even more difficult to push the gate values towards one.
>
> We would like to clarify that the gate distribution shift occurs in the first layer, and it remains when the number of layers increases from 3 to 6 as shown in Figures 5 and 6. Additionally, even in the deep layer models, gate mechanisms in the first layer do not use previous information through hidden states. We therefore argue that the shift should also occur in the deep models.
> Regarding the sequence length of WikiText-103, the number 2048 is not a whole sequence length but the context length. We have clarified this in the revised paper. Experiments on this dataset evaluate general language modeling performance rather than the ability to capture long-term dependencies.
>
> W2:
>
> Although our proposed method is simple as you have pointed out, our experiments show that our proposed method outperforms baselines in the long-sequence task. This is a particularly important practical finding, as it has been reported that linear RNNs outperforms transformers in the long-term sequence task [1].
> Our finding of the initialization dependency of gate linear RNNs is also novel and practically relevant. We expect that leveraging our work to modern gated linear RNNs like Mamba can even improve the ability to capture long-term dependencies.
>
> Q:
>
> Thank you for your comments.
> We have extended our motivation using synthetic data in the introduction section in our paper.
>
> [1] Zhen Qin, Songlin Yang, Weixuan Sun, Xuyang Shen, Dong Li, Weigao Sun, and Yiran Zhong. HGRN2: Gated linear RNNs with state expansion. In Proceedings of the Conference on Language Modeling, 2024.

---

### Official Review · Reviewer_eQho · 2025-10-31

**Soundness:** 2
**Presentation:** 2
**Contribution:** 2
**Rating:** 2
**Confidence:** 5

**Summary:**

This paper observes the changes in the distribution of forget gate values in gate linear/nonlinear RNNs before and after training on two small-scale synthetic tasks, i.e., Copying and Selective Copying (Figures 1 & 2). Specifically, the distribution shifts from a relatively broad and uniform initialization to a state where most values cluster near the two extremes, 0 or 1. Based on this, the paper proposes using the Gumbel-Softmax trick to initialize the gate values to distributions closer to these two extremes. Experimental results on different gate initialization methods show that the proposed initialization approach can accelerate performance improvement and convergence speed for the copying/selective copying tasks (Figures 4 & 6). The paper also compares the language modeling perplexity loss of different gate initialization methods on the WikiText-103 dataset (Table 1), and conducts hyperparameter fine-tuning experiments for the proposed initialization method (Figure 8).

**Strengths:**

(1) The authors observe that in the two small-scale synthetic tasks, Copying and Selective Copying, the forget gate values in gated linear/nonlinear RNN models tend to cluster near the extreme values of 0 and 1 after training, and this phenomenon is analyzed and explored to some extent.

(2) This paper proposes a method for initializing the forget gates using the Gumbel-Softmax trick, which can accelerate both the performance improvement and convergence speed of the models on these two small-scale synthetic tasks.

**Weaknesses:**

**(1) The core contribution of this paper may lack generalizability to real-world larger-scale (with parameter size not less than 1B) pre-training and multi-task reasoning scenarios.**

The gate value distributions observed only from the Copying/Selective Copying tasks may differ significantly from those in large-scale, general language modeling settings in real-world applications. As seen in Figures 1 and 2, there is a substantial proportion of forget gates close to 1 (corresponding to input gates close to 0) after training. I believe this is because in the copying/selective copying tasks, only the segments that need to be copied require memorization, while all other segments are treated as noise and can be largely forgotten. However, in practical applications, the model not only performs retrieval operations but also needs to form summarizing memories of factual information from a large number of segments. Therefore, the extreme situation where the forget gate is close to 1 and the input gate is close to 0 should not be as common as observed in copying/selective copying tasks.

**(2) Purely from the perspective of controlling gate value distribution, the initialization method proposed in this paper is not very meaningful in deep networks (e.g., 24 layers or more).**

In shallow networks (such as the 3-layer and 6-layer models discussed in this paper), the token mixing range that is achieved by gate values close to 0 or 1 could, in deeper networks, be equivalently modeled by progressively increasing the lower bound of the forget gate across layers. Mathematically, this approach (which was proposed in the HGRN paper[2]) should be able to fit the same effect.

**(3) The experimental evaluation is apparently not comprehensive enough.**

Recent studies (for example, Table 1 in the DeltaNet paper [1]) have conducted comprehensive evaluations of model performance at scales such as 340M/1.3B, across more than a dozen commonsense reasoning and retrieval tasks. The persuasiveness of these results is much higher than the language modeling experiments in this paper, which are limited to Wikitext-103.

Minor weaknesses:

(1) The phrase "removing the hidden states" may cause ambiguity; it might be better to use "decay" or "forget" instead.

References:
[1] https://arxiv.org/pdf/2406.06484
[2] https://arxiv.org/abs/2311.04823

**Questions:**

My questions have been presented in the WEAKNESSES section.

---

> ### Author Response · Authors · 2025-11-23
>
> We appreciate your thoughtful feedback.
>
> W1:
>
> One of our key findings is that, even in simple copying and selective copying tasks, gated values of gated linear RNNs fail to approach zero or one when the initialization is not appropriate.
> This is not desirable because large gate values close to one are necessary to capture long-term dependencies.
> This finding indicates that it could be more difficult to capture long-term dependencies for real-world applications because the higher complexity of real-world data makes it even more difficult to push the gated values towards one.
>
> Moreover, our proposed method is not designed solely for retrieval operations because our proposed initialization is applied to only the first layer.
> Through the second layer beyond, our proposed method can summarize memories from a large number of segments.
>
> W2:
>
> We would like to clarify that the gate distribution shift occurs in the first layer, and it remains when the number of layers increases from 3 to 6 as shown in Figures 5 and 6. This indicates that the shift should also occur in the deeper models.
> Please note that, even in the deep layer models, gate mechanisms in the first layer do not use previous information through hidden states.
> HGRN is an effective approach to increase gate values by raising lower bounds from input to output layer. However, since HGRN does not apply the lower bound to the first layer, their gated values can fail to approach one at the first layer.
> As shown in Figure 4, the lower bounds cannot achieve optimal performance when initialization does not work well at the first layer.
>
> W3:
>
> Our primary goal is to understand the effect of hidden states on the gate mechanisms, rather than providing state-of-the-art performance on benchmark tasks.
> As we have addressed in our response to W1, our findings can be even more critical for modeling real-world data because they are potentially more complex and large-scale, which makes pushing the gate values towards one more difficult.
> Our experiments on WikiText-103 verifies the versatility of our proposed method, and those on synthetic data show improvement of capturing long-term dependencies.
> Scaling to large models remains future work.
>
> Miner weakness:
>
> In this work, we focus on the difference of gate mechanisms between gated linear RNNs and non-linear RNNs. Therefore, we use the word “removing” since gated linear RNNs can be considered as gated nonlinear RNNs without hidden states in their gate mechanism, and your suggestions “decay” and “forget” are unfortunately inaccurate expressions in this context.

---

### Meta-Review · Area_Chair_jfG4 · 2025-12-31

**Summary:**

This paper investigates the gating dynamics of gated linear RNNs, focusing on the impact of removing hidden states from the gate computation. The authors propose a bias initialization method based on the Gumbel-Softmax trick. The proposed method is evaluated on synthetic and real-world sequence modeling tasks with improvements over standard and prior initialization strategies.

**Reviewer Concerns:**

1. The main claim of this paper is somewhat not very convincing, lacking generalizability to real-world, larger-scale, and multi-task reasoning scenarios.
2. The experimental evaluation is apparently not comprehensive enough.
3. Purely from the perspective of controlling gate value distribution, the initialization method proposed in this paper is not very meaningful in deep networks.

**Reviewer Scores:**

The paper received all negative ratings before the rebuttal. The rebuttal has not addressed most of the main concerns from the reviewers. The final decision is rejection.

---

### Decision · Program_Chairs · 2026-01-26

Reject